# Six years of measuring patient experiences in Belgium: Limited improvement and lack of association with improvement strategies

Astrid Van Wilder[1]*, Kris Vanhaecht[1,2], Dirk De Ridder[1,3], Bianca Cox[1],
Jonas Brouwers[1], Fien Claessens[1], Dirk De Wachter[4], Svin Deneckere[1,4],
Dirk Ramaekers[1,5], Else Tambuyzer[6], Ilse Weeghmans[6], Luk Bruyneel[1,2]

1 Leuven Institute for Healthcare Policy, KU Leuven—University of Leuven, Leuven, Flanders, Belgium,
2 Department of Quality Improvement, University Hospitals Leuven, Leuven, Flanders, Belgium,
3 Department of Urology, University Hospitals Leuven, Leuven, Flanders, Belgium, 4 Flemish Institute for
Quality of Care, Brussels, Belgium, 5 Flemish Hospital Indicator Initiative, Brussels, Belgium, 6 Flemish
Patient Platform, Brussels, Belgium

* astrid.vanwilder@kuleuven.be

pone.0241408

CANADA

**Data Availability Statement:** Data on patient
experiences for each individual Flemish hospital is
publicly available via www.zorgkwaliteit.be

## Abstract

### Objective

To examine trends in patient experiences in the period 2014–2019, describe improvement
strategies implemented by hospitals in the same period, and study associations between
patient experiences and implemented strategies.

### Design

Multi-center retrospective region-wide observational design.

### Setting

Flanders, Belgium.

### Participants

44 out of 46 Flemish acute-care hospitals publicly reporting patient experiences via the
Flemish Patient Survey (FPS).

### Main outcome measure(s)

Primary outcomes were the two global FPS ratings: percentage of patients rating the hospital 9 or 10 and percentage of patients definitely recommending the hospital. Secondary outcomes were the average top-box score percentages for each of the 8 remaining dimensions
of the FPS.

### Results

Between 2014 and 2019, there was a significant improvement in patients scoring the hospital 9 or 10 (56% to 61%) and patients definitely recommending (67% to 70%) the hospital.

Additionally, all relevant data are within the manuscript and its Supporting Information files.

**Funding:** This work was supported by the hospital umbrella organization Zorgnet-Icuro, who allocated a research chair on hospital quality to the Leuven Institute for Healthcare Policy. This research chair (grant number EZG-LSICR1-O2010) was awarded in name of Zorgnet-Icuro by Mrs. Margot Cloet to Prof. Dr. Kris Vanhaecht and Prof. De Dirk De Ridder, who are both authors of this manuscript. The funders had no role in study design, data collection and analysis, decision to publish, or preparation of the manuscript.

**Competing interests:** The authors have declared that no competing interests exist.

Significant increases in patient experiences over time were also observed in other dimensions, except for the dimension discharge. Hospital key informants reported various improvement strategies related to patient experiences with care and the FPS. Feedback to nursing wards (n = 44, 100%) and clinicians (n = 39, 89%) were most common. Overall, most improvement strategies were not or only weakly associated with patient experience ratings in 2019 and changes in ratings over time. Still, positive associations were discovered between the strategies 'nursing ward interventions' and 'hospital wide education' and recommendation of the hospital.

## Conclusions

Patient experiences have improved modestly in Flemish acute-care hospitals. Hospitals report to have invested in patient experience improvement strategies but positive associations between such strategies and FPS scores are weak, although there is potential in further exploring nursing ward interventions and hospital wide education. Hospitals should continue their efforts to improve the patient's experience, but with a more targeted approach, taking the lessons learned on the efficacy of strategies into consideration.

## Introduction

Hospitals are increasingly integrating patient-centeredness within their policy statements. Its importance as one of the dimensions of healthcare quality [1] is becoming more and more recognized. Patient-centered care is associated with improved clinical outcomes and reduced costs [1–4]. Assessing the patient's perspective of quality has long been described as a valuable quality indicator [5] and the foundation of patient-centeredness. Many health systems have therefore developed survey instruments aimed at measuring patient experiences, like the Hospital Consumer Assessment of Healthcare Providers and Systems (USA) [6] and the NHS Patient Survey (UK) [7] for acute-care hospitals. In Flanders, the northern part of Belgium, a uniform instrument was developed by the Flemish Patient Platform and validated [8] under the heading of the Flemish Patient Survey (FPS). The stakeholder-initiated Flemish Hospital Indicator Initiative (VIP²) aimed to increase insight into the quality of its hospitals by using clinical process and outcome indicators. Amongst other indicators, patient experiences with care, are voluntarily gathered hospital-wide via FPS by nearly all Flemish hospitals. In order to support quality improvement initiatives, feedback is available to all organizations. Communication of individual results on hospital websites is encouraged. In 2015, a central website (http://www.zorgkwaliteit.be) was developed where findings can be consulted by the public in an aggregated manner. The top-box scores of two global patient experience measures, i.e. patients definitely recommending the hospital and patients rating the hospital 9 or 10, are publicly reported once a year since July 2015.

Merely implementing a patient experience survey does not suffice to improve patients' experiences [9]. Reporting of patients' perspectives of hospital care can, however, be an incentive to enhance and reinforce quality improvement, although international evidence remains scant and ambiguous [10] and is often based on case studies and expert opinion [11–13]. A recent systematic review [14] looked into initiatives to improve patient satisfaction and observed potential in strategies concerning communication [15], patient [16] and physician education [17] and increasing pharmacists' involvement [18]. Making use of online platforms

like Yelp or Facebook could be linked with improvements in patient experiences [19, 20]. Aboumatar and colleagues [21] studied high-performing US hospitals of patients' reports of care and found involvement and responsibility at multiple levels of the organization, from leaders to clinicians, to be a common trait. They found that high-performing hospitals used multiple and similar concurrent interventions to improve patient experiences, like nursing ward interventions or hospital-wide feedback. External incentives like accreditation [22–24] or pay for quality in a Value Based Purchasing program [25] were found to have little impact on the patient's experience.

How patient experiences have evolved in Flanders since the first public release in July 2015 of 2014 scores, is unclear. Additionally, which quality improvement strategies concerning patient experiences have been introduced in Flemish hospitals remains unexplored. The aim of this study was to describe associations between improvement strategies and patient experiences as assessed via the FPS. We therefore first examined trends in patient experiences from 2014 to 2019. Subsequently, we described which strategies Flemish acute-care hospitals have implemented during the same time period. Finally, associations between patient experiences and improvement strategies were explored.

## Materials and methods

### Study design

A multi-center retrospective region-wide observational study.

### Study sample and recruitment

The FPS is handed out to all eligible patients (i.e. all discharged non-psychiatric patients above 18 years of age) during two periods of the year (6 weeks in March-April and 6 weeks in September-October) and with a yearly minimum of 300 filled out surveys per hospital. Over the study period, on average 78% of hospitals distribute their surveys on paper, 11.6% handed out an electronic version of the FPS and 10.4% combined electronic with paper distributions. Key informants from all Flemish acute-care hospitals (n = 55) who have chosen to publicly report (n = 46) patient experience scores on http://www.zorgkwaliteit.be were contacted for participation in this study, encouraged by the hospital umbrella organization Zorgnet-Icuro. Email and telephone reminders were sent by the research team to non-responsive hospitals.

### Data collection

To describe trends in FPS results, the Flemish Institute for the Quality of Care was contacted as the official organization overseeing the development and measurement of quality indicators. Patient-mix adjusted quality indicators, aggregated at hospital-level, were provided from the earliest collections in 2014 to the first semester of 2019 within the 'patient experiences' domain of the Flemish Indicator Initiative. This encompasses the percentages of top-box scores on 28 questions concerning nine dimensions of patient experience: hospital stay preparation, information about condition, information about treatment and procedures, dealing with patients and collaboration between healthcare providers, privacy, safe care, pain management, discharge and global experience. The two global patient experience measures, i.e. patients grading the hospital and patients recommending the hospital, are the sole indicators publicly reported online at the time of the study. Patient-mix adjustments include patient age, sex, housing type, health status and level of education.

To outline currently implemented quality improvement strategies, an online survey with personal code was sent out in summer 2019 via Qualtrics© to all quality managers within the

study sample. The survey was developed within the research team and contained 16 binary (yes/no) questions about hospital participation in strategies. The inquired strategies were based on international literature of frequently implemented initiatives aimed at improving patient experiences.

## Statistical analysis

We first described our sample characteristics. Main outcomes were the two global patient experience measures: the percentage of patients rating the hospital 9 or 10 and the percentage of patients definitely recommending the hospital. Secondary outcomes were the average top-box score percentages for each of the 8 remaining dimensions of the FPS. To describe the trend in patient experiences, our first research objective, we plotted the two global top-box measures from 2014 to 2019 for each participating hospital. Linear changes in top-box percentages over time were modelled using a separate multilevel model for each outcome, accounting for repeated measures through a random intercept for hospital. In a second set of models, year was treated as a categorical variable to allow for non-linear trends. For our second objective concerning implemented strategies, we present the findings from the survey on quality improvement initiatives visually by percentage of participating hospitals and by percentage of implemented strategies. For our final research objective, we studied the effect of improvement strategies as potential predictors of superior patient experience scores on the FPS. Using separate models for each outcome, we tested differences in percentage top-box scores measured in 2019 between hospitals with and without a specific strategy (linear regression), as well as differences in linear trends, i.e. the evolution of percentage top-box scores from 2014 to 2019 (multilevel linear regression). Differences in time trends between hospitals with and without a strategy were assessed using an interaction term between a binary indicator for strategy implementation and a linear variable for year. The strategy "FPS feedback to nursing wards" was not tested as this was implemented by all 44 hospitals. Statistical significance of the regression analyses was determined at an alpha level of 0.05. The critical threshold for the regression analyses concerning associations with implemented strategies was determined at p<0.0033, which is derived from a Bonferroni correction [26] to control for multiple testing, i.e. alpha level of 0.05 divided by 15, the number of strategies tested. The analyses for this paper were generated using SAS$^©$ software, Version 9.4 of the SAS System for Windows.

## Ethical considerations

The study protocol was approved as part of a larger retrospective observational study concerning the impact of improvement initiatives on patient outcomes by the Ethics Committee of University Hospitals Leuven (S63449).

## Results

### Sample

Our final sample included 44 (response rate: 96%) acute-care hospitals who agreed to participate. Four included hospitals were university hospitals (9%) and the number of beds ranged from 170 to 1764. Seven (16%) hospitals did not start FPS measurements until 2015. Four hospitals (9%) did not measure patient experiences for one or two study years due to reasons like hospital mergers, external accreditation or moving to another building. The total number of participants filling out their patient experience increased each year from on average 613 per hospital (SD: 360.7) in 2014 to a mean of 741 (SD: 440.4) in 2018. For all participating hospitals, this totals to a sample set of 23 549 patients in 2014 and 32 464 in 2018. For the first

semester of 2019, already 16 193 patients (on average 378 per hospital) filled out the FPS, which is in accordance with expectations.

## Trend in patient experiences

The overall and hospital-specific trends in global patient experiences are plotted in Fig 1. Overall, the percentage of patients rating the hospital 9 or 10 has steadily increased from 56% in 2014 to 61% in 2019, while the percentage definitely recommending the hospital ranged from 67% in 2014 to 70% in 2019. Some hospitals (e.g. AI, AJ, and AQ) appear to follow an upward trend, while patient experiences seem to deteriorate in e.g. AH, BE and BJ. For each hospital, both global questions appear to follow similar trends, although exceptions exist (e.g. AO, AY, BA).

S1 Table displays the yearly top-box percentages and the results of the multilevel regression models across time for the two global FPS questions and the averages for the 8 remaining FPS dimensions. Large variation in average percentage top-box scores exists between the 8 dimensions, ranging from 51% to 89% in 2014 and from 53% to 88% in 2019. Assuming linearity, a significant improvement in patient experiences was observed for the two global questions and for all dimension averages except for the dimension discharge. The estimated yearly increases in the percentage of patients rating the hospital 9 or 10 and the percentage of patients definitely recommending the hospital were 1.10 (95% CI: 0.80; 1.40) and 0.39 (95% CI: 0.15; 0.63) respectively. Results from regression models treating year as a categorical variable indicate that improvements are primarily observed in recent measurement periods: compared with 2014, a significant increase in top-box percentages was observed for 2 out of 10 outcomes in 2017, and for 8 out of 10 outcomes in 2019. The largest improvement in patients' experience was observed for the dimension safe care, with 52% of patients answering the top-box score in 2014, improving to 64% in 2019 (β = 11.69, 95% CI: 10.03; 13.34). Worsening of patient experiences could be observed in the dimension discharge. However, deteriorations are small and

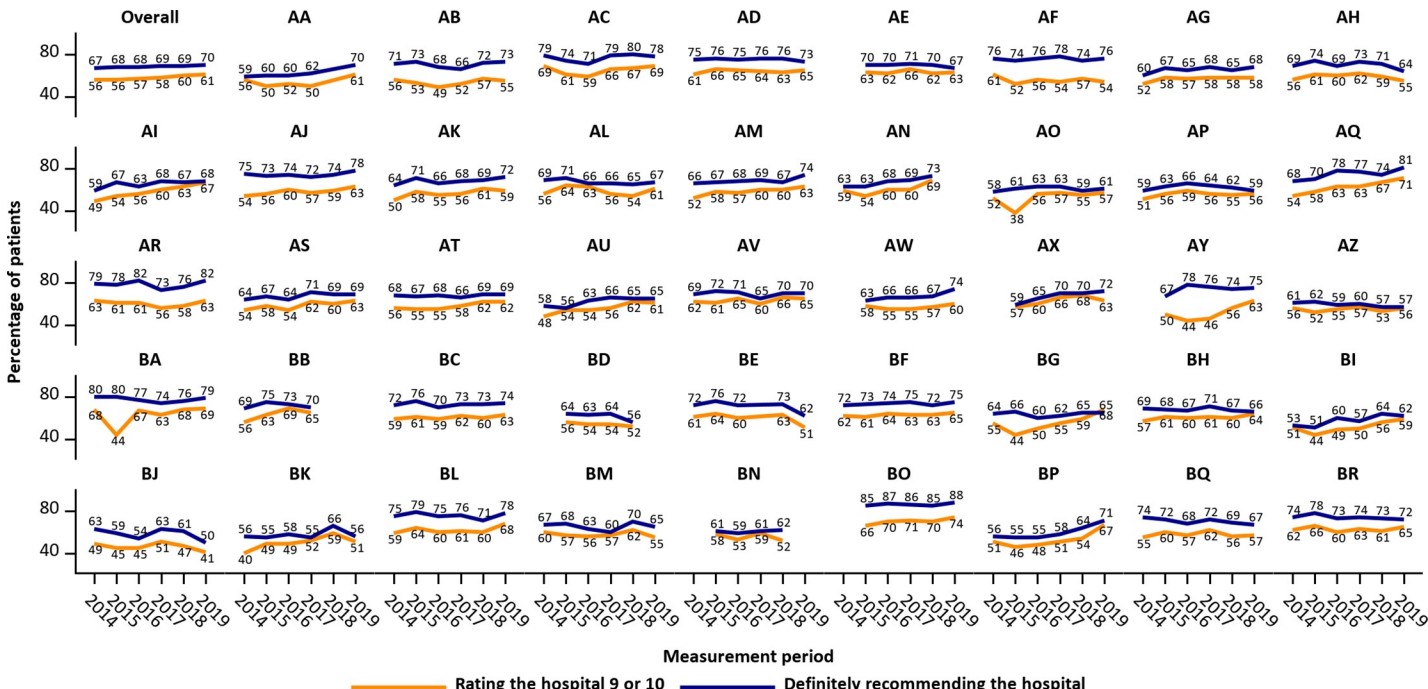

**Fig 1. Hospital trends in patient experience scores for the two global questions.** Each figure represents the percentage top-box scores in one of 44 participating Flemish acute-care hospitals. The upper left figure represents results aggregated for all participating hospitals.

scores remain high (average percentage top-box scores 89% in 2014 and 88% in 2019, β = -0.63, 95% CI: -1.19; -0.08).

## Implemented strategies to improve patient experiences

An overview of the surveyed strategies with a description of each strategy is provided in Table 1, which includes examples of strategies employed by participating hospitals. Analysis of the binary survey questions on improvement strategies resulted in the heatmap displayed in Fig 2. FPS feedback to nursing wards is a strategy implemented by all hospitals (100%, n = 44),

**Table 1. Surveyed strategies and their description.**

| Surveyed strategy | Description |
|---|---|
| FPS feedback to nursing wards | Flemish Patient Survey feedback is received by nursing wards on a regular basis. Feedback can occur on internal data collection as well as on the external benchmark reports released twice a year. |
| FPS feedback to clinicians | Flemish Patient Survey feedback is received by clinicians on a regular basis. Feedback can occur on internal data collection as well as on the external benchmark reports released twice a year. |
| Nursing ward interventions | Interventions at the level of the nursing ward are implemented to improve patient experiences. Examples include the introduction of a Magic Table© on geriatrics, interventions on pain management, organizing mealtimes between staff and patients where patients can express their concerns, or the introduction of Patient Reported Outcome Measures (PROMs) on specific wards. |
| Hospital wide interventions | Hospital wide interventions are launched to improve patient experiences. Examples are the implementation of an incident reporting system designed for patients or the organization of consultation hours between hospital staff and management and patients. Additionally, interventions could comprise hospital-wide campaigns aimed at improving the patient's experience. Examples include participation in the internationally renowned 'What Matters to You' campaign, based on Barry and Edgman-Levitan's perspective [27] or campaigns concerning Mangomoments based on research by Vanhaecht *et al.* [28]. |
| Board sets strategy | The hospital board sets the strategy to improve patient experiences. The strategy can e.g. be documented in a charter which is then distributed to all staff. |
| FPS targets | Specific targets concerning Flemish Patient Survey are premised. A hospital can e.g. choose to aim for more than the required 300 yearly surveys, or can aim for a specific percentage gain in one or more patient experience dimensions. |
| Hospital wide education | Hospital wide education, like workshops or seminars, to improve patient experiences are organized. For example, hospitals could develop a hospital academy, wherein both online and offline courses are organized for both care professionals and patients. Topics for professionals could include ways of introducing yourself to the patient and techniques on informing patients about their treatment. |
| Discharge info on admission | Discharge information is provided at the time of a patient's admission. |
| Nursing rounds | Nursing rounds specifically aiming to improve patient experiences are organized. |
| HR Policy | Improving patient experiences is an area of concern for human resources management. How an individual care provider scores on his/her patient's experience, can be a topic of a performance appraisal. |
| Proactive discharge calls | A selection of patients is called proactively after discharge. |
| Bedside briefing | Briefing of care providers at shift transfer takes place at the patient's bedside. |
| Social media follow-up | Reviews by patients on online platforms like Facebook, Twitter, Google Reviews, etc. (social media) are systematically followed up on. |
| FPS nursing ward rewards | Nursing wards receive a reward when scoring excellently on Flemish Patient Survey. The reward can be of a financial nature, but can also e.g. entail a teambuilding outing. |
| Multidisciplinary discharge | A multidisciplinary team of care providers is present at patient's discharge. |
| External consultants | A consultancy firm is hired to improve patient experience scores. |

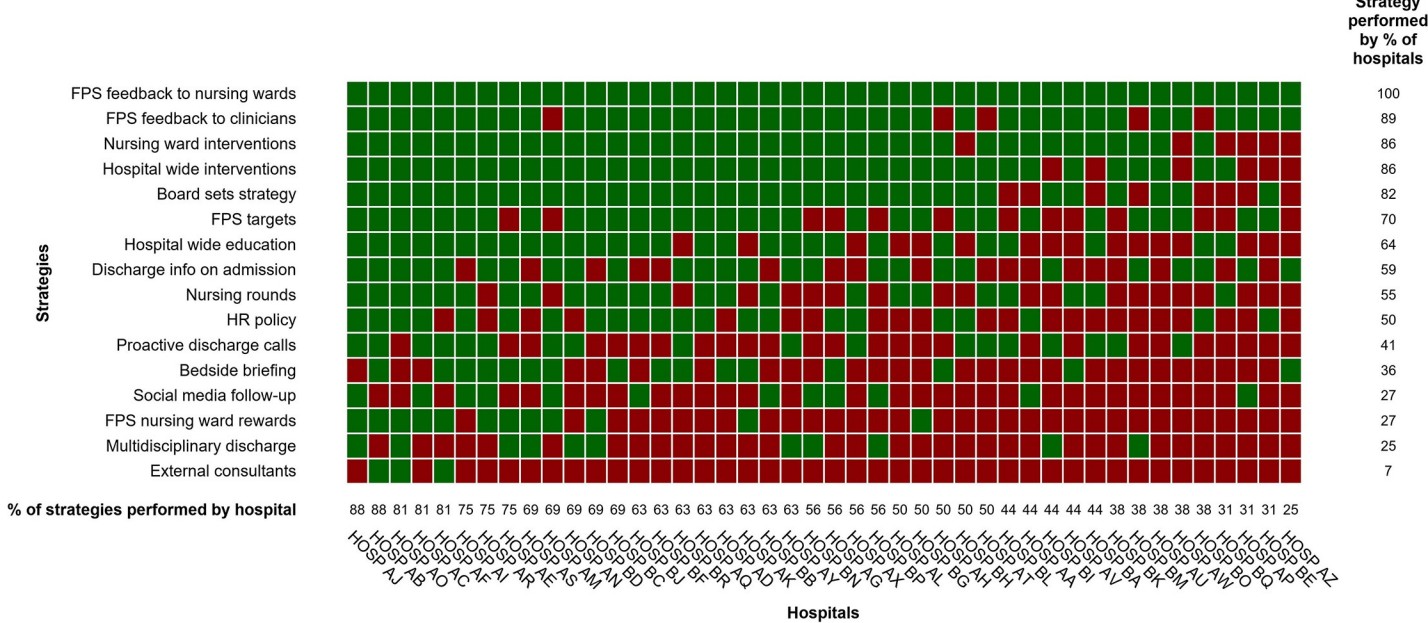

**Fig 2. Implemented quality improvement strategies to improve patient experiences across hospitals.** Each cell represents a quality improvement strategy in one particular participating hospital (n = 44). A green cell represents the strategy being implemented, whereas a red cell represents an unimplemented strategy.

while direct feedback to clinicians (89%, n = 39) is second most common. In a shared third and fourth place come nursing ward interventions (86%, n = 38) and hospital wide interventions (86%, n = 38). Conversely, hiring external consultants to improve patient experiences is the least explored strategy (7%, n = 3). Discharging the patient with a multidisciplinary team (25%, n = 11) and both rewarding the best FPS performing nursing ward (27%, n = 12) and social media follow-up (27%, n = 12) are relatively infrequent as well. A large variation between the number of strategies a hospital implements can be observed, ranging from 4 to 14 out of 16 surveyed initiatives. The number of strategies is independent of hospital size or teaching status. Among the 5 hospitals employing the most strategies for example, both academic (n = 2) and general (n = 3) hospitals are represented, which are located in 4 of the 5 Flemish provinces and with the number of beds ranging between 271 and 1049.

## Associations between patient experiences and improvement strategies

Associations between the strategies reported by the participating hospitals and the two global patient experience questions for the first semester of 2019 are displayed in Table 2. None of the strategies were associated with rating of the hospital, whereas top-box scores for recommendation of the hospital were significantly higher for hospitals having implemented nursing ward interventions and hospital wide education. For both strategies, the difference in percentage definitely recommending the hospital between hospitals with and without the strategy was around 6.6%, but these associations were not significant after Bonferroni correction. At an alpha level of 0.05, significant positive associations were observed for 6 strategy-dimension combinations (S2 Table), including 3 dimensions for the strategy nursing ward interventions and 2 dimensions for the strategy hospital wide intervention. The dimension discharge, however, was negatively associated with the strategies FPS feedback to clinicians and external consultants. The latter was also negatively associated with the dimension preparing for hospital stay. However, after Bonferroni correction, none of these associations remained significant.

**Table 2. Associations between quality improvement strategies and top-box scores for global patient experience questions in 2019.**

| Surveyed quality improvement strategy | Percentage rating the hospital 9 or 10 | | Percentage definitely recommending the hospital | |
|---|---|---|---|---|
| | β[1] | (95% CI) | β[1] | (95% CI) |
| FPS feedback to clinicians | -0.64 | (-6.61; 5.32) | -2.66 | (-9.89; 4.58) |
| Nursing ward interventions | 4.69 | (-0.64; 10.01) | 6.64 | (0.23; 13.05)* |
| Hospital wide interventions | 3.30 | (-2.13; 8.72) | 5.00 | (-1.56; 11.56) |
| Board sets strategy | -1.06 | (-5.98; 3.86) | -0.81 | (-6.83; 5.21) |
| FPS targets | -0.14 | (-4.45; 4.16) | 1.92 | (-3.31; 7.14) |
| Hospital wide education | 2.61 | (-1.34; 6.55) | 6.69 | (2.26; 11.13)** |
| Discharge info on admission | 1.03 | (-2.98; 5.05) | 3.63 | (-1.15; 8.41) |
| Nursing rounds | 2.24 | (-1.65; 6.13) | 2.45 | (-2.31; 7.21) |
| HR policy | 0.08 | (-3.87; 4.03) | 1.74 | (-3.05; 6.53) |
| Proactive discharge calls | 1.60 | (-2.36; 5.56) | 4.68 | (-0.11; 9.48) |
| Bedside briefing | -0.26 | (-4.29; 3.77) | 1.74 | (-3.15; 6.63) |
| Social media follow-up | -0.54 | (-5.09; 4.02) | 0.09 | (-5.48; 5.66) |
| FPS nursing ward rewards | 0.39 | (-4.03; 4.81) | 3.47 | (-1.81; 8.76) |
| Multidisciplinary discharge | 0.12 | (-4.82; 5.05) | -1.52 | (-7.52; 4.49) |
| External consultants | -6.48 | (-13.68; 0.72) | 0.21 | (-8.94; 9.36) |

[1] The difference (with 95% confidence interval) in percentage top-box scores between hospitals with and without the improvement strategy.

* Statistically significant at an alpha level of 0.05.

** Statistically significant at an alpha level of 0.01.

None of the estimates were significant after Bonferroni correction.

Associations between strategies and trends in top-box score percentages over time are presented in Fig 3 (two global questions) and S1 Fig (8 remaining dimensions). Significant differences in time trend slopes were observed for the strategy nursing ward interventions: top-box scores for both global questions increased over time in hospitals with nursing ward interventions, whereas patient experiences remained constant (rating the hospital) or deteriorated (recommending the hospital) in hospitals without nursing ward interventions. For recommendation of the hospital, significant differences in time trends were also observed for the strategies board sets strategy, social media follow-up, and multidisciplinary discharge, with hospitals that implemented these strategies showing more positive slopes than hospitals without the strategy. Hospital rating, however, increased more steeply in hospitals without than in hospitals with bedside briefing, but the latter started with higher scores and both ended with similar scores in 2019. Only the association between nursing ward interventions and recommendation of the hospital remained significant after Bonferroni correction. Bonferroni-corrected significant differences in time trends between hospitals with and without nursing ward interventions were also observed in the dimension dealing with patients and collaboration between healthcare providers, with patient experience scores increasing over time in hospitals with nursing ward interventions, but decreasing in hospitals without nursing ward interventions. Patient experience scores in the dimension safe care increased more steeply over time in hospitals with board setting strategy than in hospitals without this strategy (significant after Bonferroni correction).

The plotted time trends are the predictions from multilevel regression models containing a binary indicator for strategy implementation, a linear variable for year, and an interaction between these variables. The p-value represents the significance of the interaction term and indicates whether time trends are significantly different between hospitals with and without a given strategy.

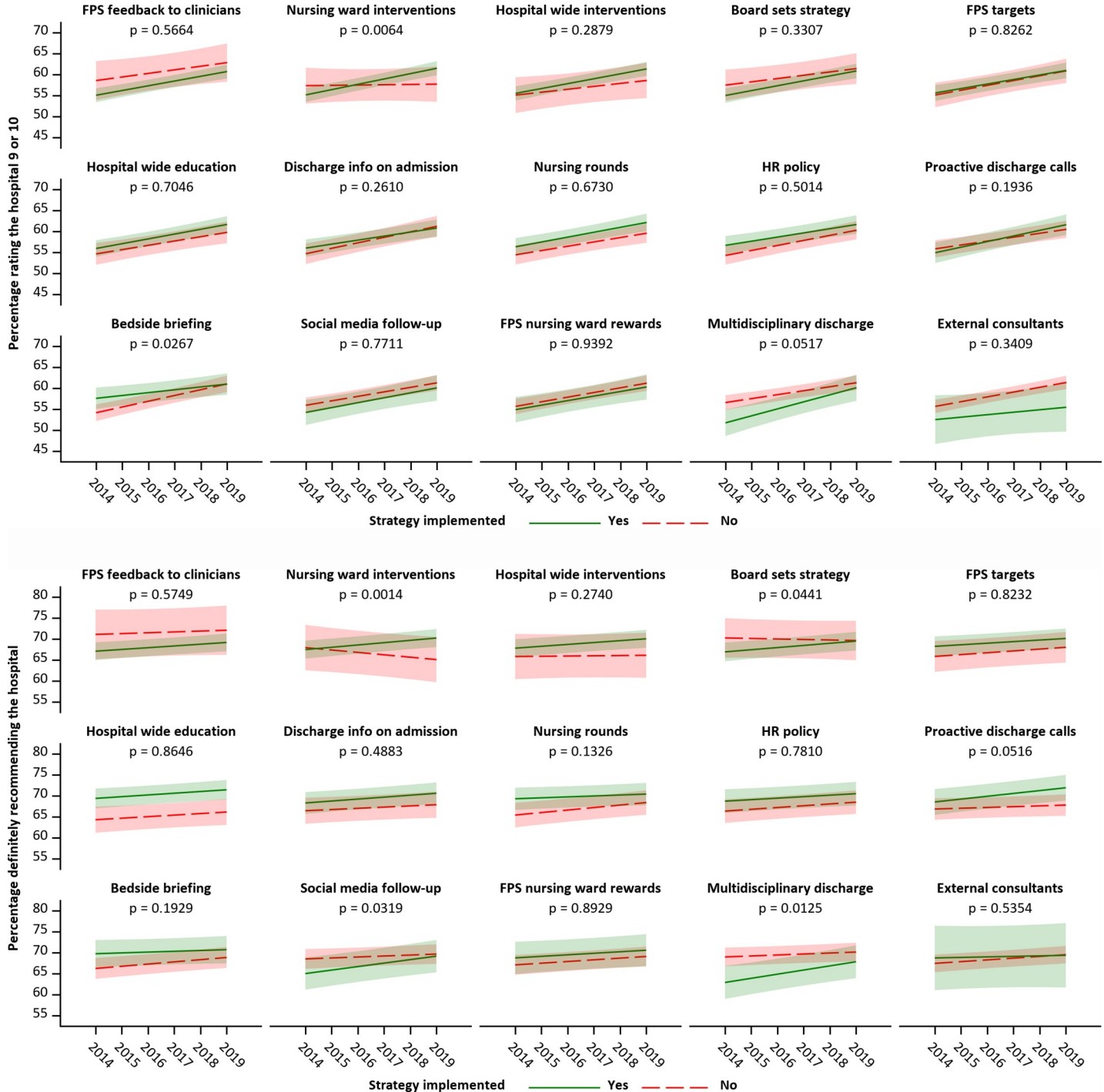

**Fig 3.** Associations between quality improvement strategies and time trends in top-box scores for global patient experience questions (upper panel: Rating the hospital; bottom panel: Recommending the hospital).

## Discussion

Although individual results of global FPS questions are already publicly reported from 2014 onwards, this paper provides the first overview of the evolution of FPS results in Flanders

across time. The overall improvement, strongest in most recent years, is commendable, yet small. The most recent top-box score of 61% of patients rating the hospital 9 or 10 e.g. is still 11 percentage points lower compared to the average of 73% in the US [29]. The percentage of patients recommending the hospital in 2019 in Flanders (70%) is still 4 percentage points removed from the current US average of 74% [29]. While one cannot unambiguously compare patient experiences across cultures and health care systems [30], the evidence seems to suggest that Flemish hospitals should keep striving for better achievements. Moreover, our study brought to light a large variability in patient experience scores across both individual hospitals and FPS dimensions. Reducing this variation has long been known as a valuable tool to improve quality of care [31]. While patient experience scores improved in 8 out of 9 dimensions, especially when concerning the safety of care, further opportunities lie in optimizing the discharge process, which seems to have stagnated over time, as well as focusing on the provision of information about both condition and treatment. The latter remain low-scoring dimensions that have shown little improvement over time. From December 2019 onwards, the website https://www.zorgkwaliteit.be has started to also publicly report specific FPS scores of all domains next to the global measures. What the impact of this public reporting on specific FPS scores will be, needs to be studied further.

As demonstrated by our survey concerning improvement strategies, Flemish hospitals have been investing modestly in improving patient experiences. While considerable variation in strategy implementation can be observed between hospitals, it is worth noting that each hospital has implemented more than one strategy. Many strategies described by Aboumatar and colleagues [21] as implemented in top-scoring US hospitals, like nursing ward interventions and hospital wide education, are also frequently implemented in Flemish hospitals. What's more, both nursing ward interventions and hospital wide education were found to be associated with better 2019 FPS results. Additionally, nursing ward interventions in particular were positively associated with improved global patient experiences over time. Flemish hospitals who did not employ nursing ward interventions scored on average 7 percentage points lower on recommendation of the hospital and even decreased across time.

To our knowledge, this is the first assessment of associations between quality improvement strategies and patient experience scores. Despite the positive associations between both nursing ward interventions and hospital wide education and 2019 FPS results and the positive relationship between nursing ward interventions and recommendation of the hospital, improvement strategies were overall not or only weakly associated with patient experience ratings. After Bonferroni correction, only the association between nursing ward interventions and improvements in recommendation remained. Additionally, the relationship with 8 specific patient dimensions is non-existent, apart from a coherent positive influence of nursing ward interventions and strategies by the board on the change in dealing with patients and provision of safe care respectively. A thorough revision of the hospitals' current approach on improving patients' experiences is therefore recommended. Considering its potential, further research into the benefits of nursing ward interventions or a hospital-wide educational program is advised. By researching the evidence-base on the interventions that have shown most promise, we hope future healthcare policy and practice might be altered towards a more unified care, instead of the wide spectrum of sometimes ineffective interventions currently implemented. The examples provided by some participating hospitals such as e.g. mealtimes between staff and patients or the development of hospital-wide courses, suggest a large variety of ways to execute strategies. We thus encourage hospitals to share and learn from both their positive and negative experiences. By focusing on both nursing ward interventions and hospital wide education, a high visibility for the patient as well as a widespread reach of all healthcare staff can be ensured.

Next to the surveyed internal strategies, the external pay-for-performance (P4P) initiative appears to have limited impact on patient experiences at first glance. Implemented In 2018, the federal P4P initiative [32] comprised an adjusted reimbursement based on high-value quality metrics like patient experiences. No strong overall improvement could be observed between FPS results in 2018 and 2019. Today, P4P solely depends on participation in the FPS and is thus not related to hospital results. Only a small portion of hospital payment is currently at stake, i.e. about 5 million on a total budget of 6.4 billion euros for acute-care hospitals. What the impact of larger payments within the P4P scheme, tied to actual FPS results, will be, needs to be studied further. Impact of external evaluations in the form of international accreditation and governmental inspection will be studied in the near future as part of a larger retrospective study of quality improvement initiatives in Flanders.

A number of considerations that merit further attention and highlight a number of limitations to this study needs to be outlined. Firstly, our study might have suffered from recall bias. Secondly, associations between strategies and FPS results need to be interpreted prudently due to multiple testing. However, using a Bonferroni correction controls for this multiplicity issue. Thirdly, we lacked specific information on the quality improvement strategies employed by participating hospitals, like implementation date and detail on how and on what wards the hospitals chose to implement their strategies. Informal conversations with participants showed this information was not always well recorded at the management level. Often due to high staff turn-over on quality departments, more detail was unavailable for a majority of participating hospitals. Fourthly, no confounding factors like e.g. employment of experience experts or other initiatives were accounted for in this study. The survey sent to every participating hospital left room to fill out additional information in an open-ended question concerning other initiatives taken. Unfortunately, only 50% of participants filled out this question, making it unusable for regression analysis. Lastly, due to the retrospective nature of this research, no causality can be established. Still, with the large representative sample of acute-care Flemish hospitals, we managed to obtain a first overview of current quality improvement strategies and how they have affected patient experience scores.

## Conclusion

This study demonstrated how patient experiences across Flemish acute-care hospitals have marginally improved and how hospitals have invested modestly in quality improvement strategies concerning patient experiences. A large variability across hospitals persists, obstructing overall improvement. Beside nursing ward interventions and hospital wide education, which was demonstrated to have potential in further improving patient experiences, no associations between employed strategies and global patient experience scores could be identified. Within the Flemish hospital landscape, the patient's experience remains an area where progress is required. Future healthcare policy will hopefully take the conclusions from this research into account and thus lead the way towards better patient care.

## Supporting information

**S1 Table. Trends in patient experience scores across Flemish acute-care hospitals (n = 44).**
(DOCX)

**S2 Table. Associations between quality improvement strategies and average top-box scores of the 8 patient experience dimensions in 2019.**
(DOCX)

**S1 File. Associations between quality improvement strategies and time trends in average top-box scores of the 8 patient experience dimensions.**
(DOCX)

## Acknowledgments

We are grateful to the Flemish Institute for Quality of Care, supported by the Flemish Patient Platform, for making available the Flemish Patient Survey data and for merging these data into a workable data set. Furthermore, we would like to express our gratitude towards Zorgnet-Icuro for their support in encouraging hospitals to participate in this study.

## Author Contributions

**Conceptualization:** Dirk De Wachter, Luk Bruyneel.

**Data curation:** Dirk De Wachter, Svin Deneckere.

**Formal analysis:** Astrid Van Wilder, Bianca Cox, Dirk De Wachter, Svin Deneckere, Luk Bruyneel.

**Funding acquisition:** Kris Vanhaecht, Dirk De Ridder.

**Investigation:** Astrid Van Wilder, Bianca Cox, Luk Bruyneel.

**Methodology:** Astrid Van Wilder, Bianca Cox, Luk Bruyneel.

**Project administration:** Astrid Van Wilder.

**Supervision:** Kris Vanhaecht, Dirk De Ridder, Luk Bruyneel.

**Visualization:** Astrid Van Wilder, Luk Bruyneel.

**Writing – original draft:** Astrid Van Wilder.

**Writing – review & editing:** Astrid Van Wilder, Kris Vanhaecht, Dirk De Ridder, Bianca Cox, Jonas Brouwers, Fien Claessens, Dirk De Wachter, Svin Deneckere, Dirk Ramaekers, Else Tambuyzer, Ilse Weeghmans, Luk Bruyneel.

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
