## [Decision Letter · Decision Letter 0]

11 Jun 2020

PONE-D-20-04886

Six years of measuring patient experiences in Belgium: limited improvement and lack of association with improvement strategies.

PLOS ONE

Dear Dr. Van Wilder,

Thank you for submitting your manuscript to PLOS ONE. After careful consideration, we feel that it has merit but does not fully meet PLOS ONE’s publication criteria as it currently stands. Therefore, we invite you to submit a revised version of the manuscript that addresses the points raised during the review process.

Reviewers have provided detailed feedback on your manuscript and the analysis of your data. Please revise as per recommendations of reviewers. In addition, I would like to emphasize the limitation of the lack of relationship between the QI strategies and the patient experience surveys. You have mentioned the same, but this needs to be made more clear. Furthermore, I would suggest that you be more specific in your recommendations for future research, policy and practice, particularly when you talk about nursing actions. 

Please submit your revised manuscript by August 10. If you will need more time than this to complete your revisions, please reply to this message or contact the journal office at plosone@plos.org. Please include the following items when submitting your revised manuscript:

We look forward to receiving your revised manuscript.

Kind regards,

Nelly Oelke

Academic Editor

PLOS ONE

Journal Requirements:

Reviewers' comments:

Reviewer's Responses to Questions

**Comments to the Author**

1. Is the manuscript technically sound, and do the data support the conclusions?

Reviewer #1: Partly

Reviewer #2: Partly

2. Has the statistical analysis been performed appropriately and rigorously? 

Reviewer #1: Yes

Reviewer #2: No

3. Have the authors made all data underlying the findings in their manuscript fully available?

Reviewer #1: Yes

Reviewer #2: Yes

4. Is the manuscript presented in an intelligible fashion and written in standard English?

Reviewer #1: Yes

Reviewer #2: Yes

5. Review Comments to the Author

Reviewer #1: The results reported on the paper are based on a national survey of Flemish hospitals, confined to hospitals which had chosen to publicly report their data (46 hospitals, not clear what proportion of the total number of hospitals this was). Data from 44 hospitals were included in the final analysis. There is no detail on how the surveys were administered in this national effort, but a minimum of 300 surveys had to be returned by patients who had been discharged from in-patient stays. Depending on the year, between 15,000 and 32,000 patient responses were available for the authors to include in their analysis.

The authors contacted quality managers in these 46 hospitals to ask what strategies had been implemented to improve the quality of patient experience. The statistical analysis is well described and appears unremarkable. However, I thought it odd that the authors looked for associations between quality improvement strategies and survey scores in 2019 – would it not have been more logical to look for an association between QI strategies and improvement in scores (i.e. from 2014 to 2019)?

Generally speaking, there was little change in patient experience during the period studied. There was considerable variation between hospitals; a few things got a bit better and a few things got worse. The overall trend was of slight improvement.

Likewise there was little relationship between scores in 2019 and QI strategies. In two domains, hiring external consultants was associated with worse scores in 2019. This rather reinforces my view that the authors have done the wrong analysis here. Surely its likely that the hospitals with the worst scores will have been most likely to hire external consultants. Indeed, the authors comment that one of the three hospitals using external consultants was a ‘strong negative outlier’. I’d have been more interested in looking at QI strategies (and their timing) in relation to changes in scores. At the very least the authors should point out this limitation – it would be even better if they redid the analysis.

The lack of improvement is well described in the paper, but hardly surprising. Unfortunately there is little information about what the hospitals actually did, i.e. the intensity of the interventions, as the data on the interventions was based solely on the answers to 16 binary questions about what strategies the hospitals employed (e.g. “Did you feed back the results to clinicians – Yes/No”). The commonest improvement strategy reported was feedback to nursing ward and clinicians and feedback on its own is well known to be a relatively ineffective strategy for quality improvement. As the authors point out, high-performing hospitals use “multiple and similar concurrent interventions to improve patient experiences”.

Reviewer #2: The authors provide results of a study of patient experiences at hospitals in Belgium from 2014-2019. They also assess associations between different strategies employed by the hospital to improve care and patient experiences. Authors find a small increase in patient satisfaction over the study period. The manuscript will be strengthened if the authors consider the following points:

1. In lines 132-136, the authors should clarify if the survey about strategies specified a time period for the implementation of those strategies as this is important in understanding the strategy analyses.

2. lines 143-144: authors should state how the repeated measures within hospitals were accounted for. (In the results, they specify the use of multilevel linear regression, but this should be stated here in the methods for the 1st objective).

3. Authors state they use multilevel linear regression when assessing the association between strategy use and patient experience outcomes, but on line 229, they say they are only analyzing results for the 1st semester of 2019. If it is only the patient experience results from the 1st half of 2019, it is not clear why multilevel linear regression is needed.

4. Did the authors check the underlying assumptions of the models? Particularly when they start analyzing individual questions, I imagine there could be some highly skewed distributions which could pose some issues with the assumptions.

5. To give a better sense of the data, the authors should provide the median and the average number of participants per hospital (and standard deviation) that filled out patient experience across the years (here I don't mean hospital level-data, but rather across the 44 hospitals, what was the mean, median, and sd in number of participants in 2014? in 2015? etc.) This will help understand the percentage level hospital data that are being analyzed.

6. In many of the analyses, the authors use year as a categorical variable. What is the justification for this? If they want to understand a general trend, would it make more sense to use time as a continuous variable (time since 2014, for example)?

7. In Tables 1, 2, Supplemental Table 1, and the text, authors should include 95% confidence intervals for the beta estimates.

8. lines 219-220: the authors state that the number of strategies is independent of hospital size or teaching status, but no data are provided to support this statement.

9. In the analyses of the association between patient experience and improvement strategies, I wonder if more rationale can be provided, especially for the analysis of the individuals questions (rather than the global questions), since it is not always clear to me why certain improvement strategies might be associated with certain questions.

10. Also in the experience/improvement strategies analysis, authors use a Bonferroni correction that accounts for the 16 strategies investigated, but not necessarily across all outcomes. I might be misunderstanding what is done here (see point 3 above), but there are many more than just 16 comparisons being made.

11. In the Discussion (line 286), the authors mention a strong negative outlier in the external consultants analysis - what happens to the findings if this outlier is removed from the analysis (to get a sense of the influence of this one observation)?

Minor points:

1. line 203: in the Figure 1 caption, I believe "on of" should be "one of"

2. lines 215-217: the given percentages do not match Figure 2 (nor calculations out of 44 hospitals), so these should be corrected.

3. lines 276-277: it is not clear which FPS outcome this line refers to. Also, no analyses are presented to demonstrate that use (or not) of a strategy was associated with improvement over time.

6. PLOS authors have the option to publish the peer review history of their article (what does this mean?). If published, this will include your full peer review and any attached files.

Reviewer #1: Yes: Martin Roland

Reviewer #2: No

---

## [Author Response · Author response to Decision Letter 0]

9 Aug 2020

Academic Editor’s Comments

1. I would like to emphasize the limitation of the lack of relationship between the QI strategies and the patient experience surveys. You have mentioned the same, but this needs to be made more clear.

Per request of the reviewers (see below), we conducted new statistical analyses on the associations between Flemish Patient Survey (FPS) top-box score percentages and surveyed quality improvement (QI) strategies. Now, we not only look at associations for the first semester of 2019, but we also study the changes in ratings over time. Additionally, for our secondary outcomes, we no longer study each individual FPS question, but we analyze the average top-box score percentages for each of the 8 remaining dimensions of the FPS. 

The new analyses could not find any (Bonferroni-corrected) associations between QI strategies and global patient experience ratings (primary outcome) in 2019, nor with changes in global ratings over time. Only the implementation of nursing ward interventions was significantly associated with an increase in the percentage of patients recommending the hospital. Our initial message narrating a lack of relationship between QI strategies and patient experience surveys is therefore now further strengthened. 

We emphasize this in the manuscript in both discussion (lines 317-322) and conclusion (lines 355-357).

2. Furthermore, I would suggest that you be more specific in your recommendations for future research, policy and practice, particularly when you talk about nursing actions. 

We added a more specific research recommendation concerning nursing ward interventions in the discussion (lines 322-326). By researching which nursing ward interventions in particular are most promising and collecting a more in-depth evidence-base, we hope that future healthcare policy and practice might be altered to provide a more unified care that can further enhance the patient’s experience. 

3. Please ensure that your manuscript meets PLOS ONE’s style requirements, including those for file naming. 

Throughout the manuscript we made formatting changes to ensure we meet PLOS ONE’s style requirements. We added e.g. indentations at the beginning of each paragraph, used correct symbols on the author page, altered some references to comply with Vancouver guidelines and named tables and figures appropriately, both within the manuscript as in the file names. 

Reviewers' comments:

Reviewer's Responses to Questions

1. Is the manuscript technically sound, and do the data support the conclusions?

Reviewer #1: Partly

Reviewer #2: Party

After reading the reviewers’ comments, we came to the realization that our manuscript would benefit from additional analyses concerning the association with change in patient experience scores between 2014 and 2019. Additionally, as mentioned by reviewer #2, our initial analyses investigating associations between individual patient survey questions and employed strategies included some irrelevant strategy-question combinations for which no association would be expected. In the new analyses, we grouped the individual questions into 9 dimensions of patient care and we analyzed dimension-specific average scores, thereby reducing the number of statistical tests and overcoming the problem of irrelevant strategy-question combinations, resulting in more sound conclusions. 

2. Has the statistical analysis been performed appropriately and rigorously?

Reviewer #1: Yes

Reviewer #2: No

We agree with the remarks given by Reviewer #2 concerning our original statistical analysis. Shortcomings have been solved (see section on statistical analysis, lines 142-166) and will be further elucidated down below. We also noticed a small error in our original manuscript, which stated that a Bonferroni correction was applied on the 16 tested QI strategies. However, 1 strategy was employed by all hospitals, meaning it was never included in the testing of strategies (see lines 159-160). We therefore corrected our mistake in lines 163-164. 

3. Have the authors made all data underlying the findings in their manuscript fully available?

Reviewer #1: Yes

Reviewer #2: Yes

Like we did in our original manuscript, we have made all data available in either the revised document or in supplemental files. 

4. Is the manuscript presented in an intelligible fashion and written in standard English?

Reviewer #1: Yes

Reviewer #2: Yes

The revised manuscript was proofread in detail by both AVW, BC and LB to ensure a correct usage of standard English. 

Reviewers’ Comments

REVIEWER #1

1. The results reported on the paper are based on a national survey of Flemish hospitals, confined to hospitals which had chosen to publicly report their data (46 hospitals, not clear what proportion of the total number of hospitals this was). Data from 44 hospitals were included in the final analysis. 

The region of Flanders has 55 acute-care hospitals, of which 46 hospitals publicly reported their patient experiences at the time of the study. The section ‘study sample and recruitment’ was amended in line 117 to make clear what proportion of the total number of hospitals (83.6%) were reached out to for the purpose of this study. 

2. There is no detail on how the surveys were administered in this national effort, but a minimum of 300 surveys had to be returned by patients who had been discharged from in-patient stays. Depending on the year, between 15,000 and 32,000 patient responses were available for the authors to include in their analysis.

We provided further detail on how the surveys were administered in lines 114-116 of the section ‘study sample and recruitment’. As displayed in the table below, the majority of hospitals publicly reporting their patient experiences distributed the Flemish Patient Survey to their patients on paper. As the percentages of distributions fluctuate across time (no clear increase or decrease could be observed), we opted to report on the average distribution percentage across the study years. 

Year Distribution on paper Electronic distribution Mixed distribution

2014 78% 9,8% 12,2%

2015 79,2% 8,3% 12,5%

2016 83,3% 12,5% 4,2%

2017 76,5% 17,6% 5,9%

2018 80% 10% 10%

2019 71,2% 11,5% 17,3%

Additionally, we provided more information on the study sample, as was also requested by Reviewer #2 (see below for more detail). The additional information on number of patient responses per hospital, is added in the ‘Sample’ section of the results in lines 178-182.

3. The authors contacted quality managers in these 46 hospitals to ask what strategies had been implemented to improve the quality of patient experience. The statistical analysis is well described and appears unremarkable. However, I thought it odd that the authors looked for associations between quality improvement strategies and survey scores in 2019 – would it not have been more logical to look for an association between QI strategies and improvement in scores (i.e. from 2014 to 2019)? 

Generally speaking, there was little change in patient experience during the period studied. There was considerable variation between hospitals; a few things got a bit better and a few things got worse. The overall trend was of slight improvement.

Likewise there was little relationship between scores in 2019 and QI strategies. In two domains, hiring external consultants was associated with worse scores in 2019. This rather reinforces my view that the authors have done the wrong analysis here. Surely, it’s likely that the hospitals with the worst scores will have been most likely to hire external consultants. Indeed, the authors comment that one of the three hospitals using external consultants was a ‘strong negative outlier’. I’d have been more interested in looking at QI strategies (and their timing) in relation to changes in scores. At the very least the authors should point out this limitation – it would be even better if they redid the analysis.

The reason why we originally only looked at association with scores in 2019 is because of the limitation that we do not know the date of implementation of the strategies (so looking at changes in scores from 2014 onwards might not be meaningful for strategies implemented only at the end of the study period). Despite this shortcoming, we agree that investigating changes in scores over time might provide additional insight. Therefore, we additionally investigated differences in time trends between hospitals with and without a strategy, by using multilevel models including an interaction term between a binary indicator for strategy implementation and a linear variable for year. 

This analysis showed that the percentage of patients recommending the hospital increased over time in hospitals with nursing ward interventions, but deteriorated in hospitals without nursing ward interventions. A similar (Bonferroni-corrected significant) pattern was observed for the dimension dealing with patients and collaboration between healthcare providers. Finally, patient experience scores in the dimension safe care increased more steeply over time in hospitals with board setting strategy than in hospitals without.

4. The lack of improvement is well described in the paper, but hardly surprising. Unfortunately, there is little information about what the hospitals actually did, i.e. the intensity of the interventions, as the data on the interventions was based solely on the answers to 16 binary questions about what strategies the hospitals employed (e.g. “Did you feed back the results to clinicians – Yes/No”). The commonest improvement strategy reported was feedback to nursing ward and clinicians and feedback on its own is well known to be a relatively ineffective strategy for quality improvement. As the authors point out, high-performing hospitals use “multiple and similar concurrent interventions to improve patient experiences”.

As we outline in the limitation section of our revised manuscript (lines 340-344), we lacked specific information on the QI strategies employed by the participating hospitals. We e.g. had no information on the implementation date of the surveyed strategies and we lacked detail on how and on what wards the hospitals chose to implement their strategies. Informal talks with a few of the respondents of the participating hospitals, taught us detailed information on quality improvement initiatives was not always well recorded at the level of quality management departments of the hospitals. Moreover, quality management is characterized by a high turn-over of staff, leading to information being unavailable for a majority of participants.

However, about half of the participants provided us with more detail on some of the surveyed strategies as well as on other employed initiatives within their hospital in the open-ended questions of the survey. While we could not use them in the regression analyses, some examples provided were used in the newly-added Table 1 (line 229), which gives a description of each surveyed strategy. In adding this, we hope to have given you some idea on what the hospitals actually did in terms of interventions. 

REVIEWER #2

1. In lines 132-136, the authors should clarify if the survey about strategies specified a time period for the implementation of those strategies as this is important in understanding the strategy analyses.

Like stated above, we outline in the limitation section of our revised manuscript (lines 340-344), we lacked specific information on the QI strategies employed by the participating hospitals. We e.g. had no information on the implementation date of the surveyed strategies and we lacked detail on how and on what wards the hospitals chose to implement their strategies. Informal talks with a few of the respondents of the participating hospitals, taught us detailed information on quality improvement initiatives was not always well recorded at the level of quality management departments of the hospitals. Moreover, quality management is characterized by a high turn-over of staff, leading to information being unavailable for a majority of participants.

2. Lines 143-144: authors should state how the repeated measures within hospitals were accounted for. (In the results, they specify the use of multilevel linear regression, but this should be stated here in the methods for the 1st objective).

We amended the statistical analysis section in lines 147-149, where we further state how repeated measures within hospitals were accounted for. Herein, we state: Linear changes in top-box percentages over time were modelled using a separate multilevel model for each outcome, accounting for repeated measures through a random intercept for hospital. 

3. Authors state they use multilevel linear regression when assessing the association between strategy use and patient experience outcomes, but on line 229, they say they are only analyzing results for the 1st semester of 2019. If it is only the patient experience results from the 1st half of 2019, it is not clear why multilevel linear regression is needed.

We are grateful for this comment, as it pointed out an error in the original manuscript. Indeed, the analysis of results from the 1st half of 2019 did not include repeated measurement, so normal instead of multilevel linear regression was used. 

4. Did the authors check the underlying assumptions of the models? Particularly when they start analyzing individual questions, I imagine there could be some highly skewed distributions which could pose some issues with the assumptions.

Firstly, we no longer analyze individual questions in the revised version of the manuscript, as -per your suggestion- below the link between strategies and individual questions is not always clear. Instead, we now consider the average top-box score percentages within 8 Flemish Patient Survey dimensions as secondary outcomes. Top-box percentages for the two global questions as well as dimension-specific averages were relatively normally distributed. Model diagnostics were checked are were quite acceptable. 

5. To give a better sense of the data, the authors should provide the median and the average number of participants per hospital (and standard deviation) that filled out patient experience across the years (here I don't mean hospital level-data, but rather across the 44 hospitals, what was the mean, median, and sd in number of participants in 2014? in 2015? etc.) This will help understand the percentage level hospital data that are being analyzed.

We provided the average and median number of participants that filled out patient experiences across the 44 participating hospitals in lines 178 to 182 of the revised document. We described how the average has evolved between 2014 to 2018 and 2019. For full disclosure, you can find the descriptives for each study year in the table below. 

Year Median Mean Std Dev

2014 520,0 613,2 360,7

2015 596,5 626,8 308,6

2016 566,5 649,7 350,6

2017 650,0 737,3 398,8

2018 648,0 741,4 440,4

2019 384,5 379,7 195,0

6. In many of the analyses, the authors use year as a categorical variable. What is the justification for this? If they want to understand a general trend, would it make more sense to use time as a continuous variable (time since 2014, for example)?

When looking at time trends in FPS scores, we originally treated year as a categorical variable to be able to pick up potential deviations from linearity. We agree, however, that for the purpose of drawing conclusions on general trends, treating year as a continuous variable makes more sense, especially now we have added the slope analyses to assess associations with strategies, which also assumes trends to be linear. Therefore, we added the linear estimate for year as the main result to Table 1 (Supplementary table 1 in the revised manuscript), but we kept the estimates for year as categorical variable (along with the crude percentages by year), as these demonstrate that improvements in FPS scores are most pronounced during the last few years of the study period.

7. In Tables 1, 2, Supplemental Table 1, and the text, authors should include 95% confidence intervals for the beta estimates.

The newly revised manuscript contains beta estimates in Table 2, Supplemental Table 1 and Supplemental Table 2. We always provide 95% confidence intervals per reported estimate. 

8. Lines 219-220: the authors state that the number of strategies is independent of hospital size or teaching status, but no data are provided to support this statement.

We have provided additional information to support our statement concerning number of employed strategies in the revised manuscript (lines 223-226). Herein, we took a subsample of the 5 hospitals with the highest number of QI strategies implemented and saw their hospital characteristics varied largely. 

9. In the analyses of the association between patient experience and improvement strategies, I wonder if more rationale can be provided, especially for the analysis of the individuals questions (rather than the global questions), since it is not always clear to me why certain improvement strategies might be associated with certain questions.

We fully agree with your comment concerning the unclear relationship between strategies and individual patient survey questions. We therefore no longer analyze individual questions in the revised manuscript. Instead, we now consider the average top-box score percentages of 8 Flemish Patient Survey dimensions as our secondary outcomes. As discussed in lines 320 to 322 of the revised manuscript, our new analyses (looking at changes in time trends) found significant associations between nursing ward interventions and the dimension ‘dealing with patients and collaboration between healthcare providers’ as well as between the board setting strategy and the dimension ‘safe care’. Both associations can be described as logical, considering the high visibility of nursing ward interventions for the patient, as well as the impact of an integrated approach on safety of care respectively. 

10. Also, in the experience/improvement strategies analysis, authors use a Bonferroni correction that accounts for the 16 strategies investigated, but not necessarily across all outcomes. I might be misunderstanding what is done here (see point 3 above), but there are many more than just 16 comparisons being made.

Firstly, as stated above, we needed to correct our manuscript (see lines 159-164), as our Bonferroni correction only took 15 instead of 16 strategies into account. We understand the reviewer’s concern towards the multiple testing of several strategies on several outcomes, but we feel our revised manuscript already reduced the risk of false significant results, as we now no longer test for each individual survey question but for patient dimensions (n=8) only. By using a Bonferroni correction, we already apply the most conservative method, as Bonferroni does not take the correlation between outcomes into account. We therefore opted to not further take the tested dimensions into account for the Bonferroni correction. Moreover, although few associations remain significant after Bonferroni correction, our general conclusion is that associations between implemented strategies and FPS scores are weak or non-existing.

11. In the Discussion (line 286), the authors mention a strong negative outlier in the external consultants analysis - what happens to the findings if this outlier is removed from the analysis (to get a sense of the influence of this one observation)?

Our new analysis used a more recent version of the Flemish Patient Survey data, with slight changes in percentage top-box scores. This has led to subtle changes in estimates obtained for the analysis investigating associations between strategies and FPS scores for 2019. The association between external consultancy use and percentage of patients rating the hospital 9 or 10 is no longer significant now, although the estimate (-6.48; 95% CI: -13.68; 0.72) is similar to the previous one (-7.6, p=0.0409). As there are no significant associations observed for the external consultants strategy in the new results, we removed the part on external consultants from the discussion. It should be noted that results for this strategy should be interpreted with caution as only 3 hospital have implemented this strategy, resulting in wide confidence intervals (as can be seen from figure 3).

Minor points: 

1. Line 203: in the Figure 1 caption, I believe "on of" should be "one of".

This is corrected and is now stated in line 210 of the revised document. 

2. Lines 215-217: the given percentages do not match Figure 2 (nor calculations out of 44 hospitals), so these should be corrected.

The section on implemented strategies to improve patient experiences was revised and corrected. Amendments can be found in lines 217-223 of the revised document. 

3. Lines 276-277: it is not clear which FPS outcome this line refers to. Also, no analyses are presented to demonstrate that use (or not) of a strategy was associated with improvement over time.

We provided more clarity in lines 310-313 of the revised document. Additionally, as we have stated above, we added analyses to demonstrate if use of a strategy was associated with improvements over time.

---

## [Decision Letter · Decision Letter 1]

8 Sep 2020

PONE-D-20-04886R1

Six years of measuring patient experiences in Belgium: limited improvement and lack of association with improvement strategies.

PLOS ONE

Dear Dr. ASTRID VAN WILDER:

Thank you for submitting your manuscript to PLOS ONE. After careful consideration, we feel that it has merit but does not fully meet PLOS ONE’s publication criteria as it currently stands. Therefore, we invite you to submit a revised version of the manuscript that addresses the points raised during the review process.

Please revise as per reviewer one's comments. As per my suggestion in the previous comments, please provide additional information on your recommendations, particularly on those interventions that had a positive impact on patient experience - "Nursing ward interventions" and "Hospital wide education." Also, as recommended by reviewer one, please add this information on the positive interventions to the abstract and revise the final sentence of the abstract to reflect a more scholarly approach. 

We look forward to receiving your revised manuscript.

Kind regards,

Nelly Oelke

Academic Editor

PLOS ONE

Reviewers' comments:

Reviewer's Responses to Questions

**Comments to the Author**

1. If the authors have adequately addressed your comments raised in a previous round of review and you feel that this manuscript is now acceptable for publication, you may indicate that here to bypass the “Comments to the Author” section, enter your conflict of interest statement in the “Confidential to Editor” section, and submit your "Accept" recommendation.

Reviewer #1: (No Response)

Reviewer #2: All comments have been addressed

2. Is the manuscript technically sound, and do the data support the conclusions?

Reviewer #1: Yes

Reviewer #2: (No Response)

3. Has the statistical analysis been performed appropriately and rigorously? 

Reviewer #1: Yes

Reviewer #2: (No Response)

4. Have the authors made all data underlying the findings in their manuscript fully available?

Reviewer #1: Yes

Reviewer #2: (No Response)

5. Is the manuscript presented in an intelligible fashion and written in standard English?

Reviewer #1: Yes

Reviewer #2: (No Response)

6. Review Comments to the Author

Reviewer #1: The manuscript is improved from the previous version and the comments I made have been satisfactorily addressed. The only comment I would now add is that more could be made of the two strategies that appear to be associated with improvement, i.e. 'Nursing ward interventions' and 'Hospital wide education' as these are the two practical things that a reader could take from the paper. I would make two suggestions. First, somewhere in the discussion (or an appendix if necessary), the paper should describe what these actually are, to the extent that the information is available. Second, I would recommend that these two findings (almost the only positive findings in the paper) should be included in the abstract. I note that the word limit for an abstract in PLOS ONE is 300, whereas the current word count of the abstract is 246, so they have some space.

Personally, I don't like the last sentence of the abstract which is expressed in rather unscientific language. I also don't know if it's overstated - e.g. in relation to the world literature on quality improvement, is a 5% increase in top scoring hospitals in five years that bad? This could also be addressed in the discussion.

Reviewer #2: (No Response)

7. PLOS authors have the option to publish the peer review history of their article (what does this mean?). If published, this will include your full peer review and any attached files.

Reviewer #1: **Yes: **Martin Roland

Reviewer #2: No

---

## [Author Response · Author response to Decision Letter 1]

8 Oct 2020

Dear Sir, 

Dear Madam,

We are once again incredibly grateful for the thorough synthesis and review of our manuscript by the PLOS ONE editor and external peer reviewers. We are glad our previous changes have overall been well received and feel the additional comments made in this review have merit and will add to an improved manuscript. Outlined below you will find our point-by-point response to all comments to demonstrate how we took them to heart and adjusted our manuscript. The line numbers refer to the revised manuscript document without tracked changes. 

Academic Editor’s Comments

1. Please revise as per reviewer one’s comment.

See reply below.

2. As per my suggestion in the previous comments, please provide additional information on your recommendations, particularly on those interventions that had a positive impact on patient experience – “Nursing ward interventions” and “Hospital wide education.”

We added additional information on recommendations for practice in lines 328 to 336 and stated the following from line 332: “The examples provided by some participating hospitals such as e.g. mealtimes between staff and patients or the development of hospital-wide courses, suggest a large variety of ways to execute strategies. We thus encourage hospitals to share and learn from both their positive and negative experiences. By focusing on both nursing ward interventions and hospital wide education, a high visibility for the patient as well as a widespread reach of all healthcare staff can be ensured.”

We hope this information can be applied by hospitals reading this manuscript that are aiming to improve their patients’ experiences.

3. Also, as recommended by reviewer one, please add this information on the positive interventions to the abstract.

We added information on the positive associations between improvement strategies and patient experiences both in the results and conclusion section of the abstract. As such, we wrote the following in line 59: “Still, positive associations were discovered between the strategies ‘nursing ward interventions’ and ‘hospital wide education’ and recommendation of the hospital.” Additionally, in line 62, we wrote: “Hospitals report to have invested in patient experience improvement strategies but positive associations between such strategies and FPS scores are weak, although there is potential in further exploring nursing ward interventions and hospital wide education.” 

4. Revise the final sentence of the abstract to reflect a more scholarly approach.

We altered the final sentence in our abstract to “Hospitals should continue their efforts to improve the patient’s experience, but with a more targeted approach, taking the lessons learned on the efficacy of strategies into consideration.” (line 65). As such, we hope to have provided a more nuanced conclusion that takes the improvements already made into account, but also highlights the fact that most strategies have not been associated with improvements in patient experience scores. A further focus on strategies that could potentially benefit the patient’s experience is therefore required. 

Reviewers' comments:

REVIEWER #1

1. The only comment I would now add is that more could be made of the two strategies that appear to be associated with improvement, i.e. ‘Nursing ward interventions’ and ‘Hospital wide education’ as these are the two practical things that a reader could take from the paper. I would make two suggestions. First, somewhere in the discussion (or an appendix if necessary), the paper should describe what these actually are, to the extent that the information is available. 

As we mentioned in our previous review, as well as in our limitation section (line 351) of our discussion, detailed information on when, how and on what ward the hospitals chose to implement their employed strategies, is unavailable for the majority of participating hospitals. Nevertheless, we understand the need for more detailed information on the surveyed strategies, in particular those with potential benefit towards the patient’s experience. We therefore contacted the quality managers of three participating hospitals that all indicated they employed both nursing ward interventions and hospital wide education and asked them to elaborate on how they employed these strategies. While exact details could not be provided, they could give examples of interventions they had implemented, which add to the understanding of what that particular strategy entails. As such, we added more information in the description section of Table 1 (line 234) for both nursing ward interventions, hospital wide interventions and hospital wide education. We provided extra information in the discussion section as well in line 332: “The examples provided by some participating hospitals such as e.g. mealtimes between staff and patients or the development of hospital-wide courses, suggest a large variety of ways to execute strategies. We thus encourage hospitals to share and learn from both their positive and negative experiences. By focusing on both nursing ward interventions and hospital wide education, a high visibility for the patient as well as a widespread reach of all healthcare staff can be ensured.” 

In addition, we put more emphasis on the potential benefits of nursing ward interventions and hospital wide education throughout the discussion section. First, we added lines 315 to 318: “What’s more, both nursing ward interventions and hospital wide education were found to be associated with better 2019 FPS results. Additionally, nursing ward interventions in particular were positively associated with improved global patient experiences over time.” Second, we wrote lines 321-324: “Despite the positive associations between both nursing ward interventions and hospital wide education and 2019 FPS results and the positive relationship between nursing ward interventions and recommendation of the hospital, improvement strategies were overall not or only weakly associated with patient experience ratings.” Last, we made some considerations for practice in lines 328-336: “Considering its potential, further research into the benefits of nursing ward interventions or a hospital-wide educational program is advised. By researching the evidence-base on the interventions that have shown most promise, we hope future healthcare policy and practice might be altered towards a more unified care, instead of the wide spectrum of sometimes ineffective interventions currently implemented. The examples provided by some participating hospitals such as e.g. mealtimes between staff and patients or the development of hospital-wide courses, suggest a large variety of ways to execute strategies. We thus encourage hospitals to share and learn from both their positive and negative experiences. By focusing on both nursing ward interventions and hospital wide education, a high visibility for the patient as well as a widespread reach of all healthcare staff can be ensured.”

By the best of our abilities, we thus hope to have provided you with a deeper understanding of how the surveyed strategies could be implemented in practice.

2. Second, I would recommend that these two findings (almost the only positive findings in the paper) should be included in the abstract.

We added information on the positive associations between improvement strategies and patient experiences both in the results and conclusion section of the abstract. As such, we wrote the following in line 59: “Still, positive associations were discovered between the strategies ‘nursing ward interventions’ and ‘hospital wide education’ and recommendation of the hospital.” Additionally, in line 62, we wrote: “Hospitals report to have invested in patient experience improvement strategies but positive associations between such strategies and FPS scores are weak, although there is potential in further exploring nursing ward interventions and hospital wide education.” 

3. Personally, I don’t like the last sentence of the abstract which is expressed in rather unscientific language. I also don’t know if it’s overstated – e.g. in relation to the world literature on quality improvement, is a 5% increase in top scoring hospitals in five years that bad? This could also be addressed in the discussion. 

We altered the final sentence in our abstract to “Hospitals should continue their efforts to improve the patient’s experience, but with a more targeted approach, taking the lessons learned on the efficacy of strategies into consideration.” (line 65). As such, we hope to have provided a more nuanced conclusion that takes the improvements already made into account, but also highlights the fact that most strategies have not been associated with improvements in patient experience scores. A further focus on strategies that could potentially benefit the patient’s experience is therefore required. 

Lastly, to further nuance the improvements made by the participating hospitals, we removed the comment ‘yet small’ (talking about significant improvement) from the results section (line 53) of the abstract. As you rightly pointed out, a 5% increase over the course of 5 years is still a commendable achievement. We also highlight this in line 296 of the discussion of the manuscript: “The overall improvement, strongest in most recent years, is commendable, yet small.” Still, like our discussion continues, we believe there is still room for improvement, considering the achievements made in other countries and considering the fact most of the surveyed strategies had no impact on the patient’s experience.

We would like to thank you again for considering our manuscript for publication in your journal.

Yours sincerely,

Astrid Van Wilder

---

## [Editor Report · Decision Letter 2]

15 Oct 2020

Six years of measuring patient experiences in Belgium: limited improvement and lack of association with improvement strategies.

PONE-D-20-04886R2

Dear Dr. Van Wilder,

We’re pleased to inform you that your manuscript has been judged scientifically suitable for publication and will be formally accepted for publication once it meets all outstanding technical requirements.

Kind regards,

Nelly Oelke

Academic Editor

PLOS ONE
---

## [Editor Report · Acceptance letter]

20 Oct 2020

PONE-D-20-04886R2 

Six years of measuring patient experiences in Belgium: limited improvement and lack of association with improvement strategies. 

Dear Dr. Van Wilder:

I'm pleased to inform you that your manuscript has been deemed suitable for publication in PLOS ONE. Congratulations! Your manuscript is now with our production department. 

Kind regards, 

on behalf of

Dr. Nelly Oelke 

Academic Editor

PLOS ONE